# In-stream habitat availability for river dolphins in response to flow: Use of ecological integrity to manage river flows

**Shambhu Paudel** [1,2]*, **John L. Koprowski**[1,3], **Usha Thakuri**[2], **Ajay Karki**[4]

**1** School of Natural Resources and the Environment, University of Arizona, Tucson, Arizona, United States of America, **2** Tribhuvan University, Institute of Forestry, Pokhara, Nepal, **3** Haub School of Environment and Natural Resources, University of Wyoming, Bim Kendall House, Laramie, WY, United States of America, **4** Ministry of Forests and Environment, Government of Nepal, Kathmandu, Nepal

* oasis.excurrent@gmail.com

**Data Availability Statement:** All relevant data are within the manuscript.

**Funding:** The author (SP) received field research grants from: Rufford Foundation (UK),

## Abstract

Population decline and extinction risk of river dolphins are primarily associated with flow alteration. Previous studies predominantly highlighted maintenance of adequate flow for low water seasons when habitats contract and the risk of local extinction escalates. Although river dolphins are sensitive to reduction in river flow, no studies quantify the relationships between flow and ecology of river dolphins to mitigate the potential adverse impacts of flow alteration. We quantify the relationships between flow and the ecology of river cetaceans concerning Ganges River dolphins (GRD; *Platanista gangetica gangetica*) usable area availability (AWS) for the low water season at wider flows (50–575 m³/s) at finer spatial and temporal scales. This study reveals that distribution of area usable to GRD is highly regulated by the adequate flow and river attributes (velocity and depth) interactions that likely offer energetically efficient modes of locomotion to GRD, suggesting the hydro-physical environment as a major determinant of river dolphin distribution and abundance. Flow and AWS relationships indicate that the flow during the dry season negatively contributed to AWS, whereas that of pre-monsoon maximized the AWS, suggesting that modifying flow regimes does alter in-stream habitats at varying spatial scales and may influence life-history strategies. Substantial fragmentation in suitable pool availability and loss of longitudinal connectivity exhibited by dry season flow suggested a higher risk of adverse biological effects during the dry season, which may reduce population viability by reducing survivorship and reproduction failure. Owing to river dolphins' dependence on the attribute of freshwater flow, they can be expected to be more affected by flow regulations as interactive effects. Considering the seasonal effects and changes in the availability of usable areas by flow alteration, adopting effective habitat retention plans by water-based development projects appears critical to avoid further ecological risks in aquatic species conservation. Identifying priority riverscapes for river cetaceans and prioritizing investment opportunities is an essential first step towards effective riverine cetacean conservation.

Commonwealth Scientific and Industrial Research Organization (CSIRO-Australia), and WWF-Nepal. The funders had no role in study design, data collection and analysis, decision to publish, or preparation of the manuscript.

**Competing interests:** The authors have declared that no competing interests exist.

## Introduction

As the human population size increases, freshwater species and ecosystems are increasingly threatened by many development activities, including habitat alteration, river diversions, fragmentation and flow regulation, expansion of agricultural and urban landscapes, and climate change [1]. All these anthropogenic consequences determine the quality and quantity of freshwater, which affects ecological processes and dynamics that determine freshwater ecosystem productivity and integrity. As withdrawals and diversions are the main threats to freshwater availability and quality [2–4], freshwater biodiversity is primarily threatened by habitat fragmentation and flow regulation [5]. Worldwide, agriculture accounts for about 70% of all freshwater usage, compared to 20% and 10% by industry and domestic use, respectively [6]. Accelerated water extraction or diversion combined with changes in climate, human population, and changing land use puts immense pressure on river basin biodiversity and ecosystem services through habitat loss and fragmentation [7].

Globally, a sharp decline in freshwater species, especially megafauna with large body size (adult body weight ≥30 kg) and complex habitat requirements, is commonly attributed to water diversion and extraction. From 1970 to 2012, populations of vertebrate freshwater declined by 88% due to habitat degradation induced by flow regulation [8, 9]. Several river basins in Southern Asia (Yangtze River, Ganges River, Indus River, Brahmaputra River) and Northern South America (Amazon, Orinoco, and Tocantins-Araguaia) have been impaired by large water extraction projects (i.e., hydropower and agriculture irrigation projects) with negative impacts on aquatic species conservation [1, 10, 11]. Such mega-projects threaten freshwater species with isolation, reduction in abundance and range, and extinction, especially in the case of apex predators and highly migratory river dolphins [10–13]. As river cetaceans are sensitive to river flow, reduced in-stream habitat quality due to the lack of adequate river flow further escalated endangerment and extinction risk by increasing entanglement and adverse biological effects through fisheries-freshwater cetaceans interactions [14]. Risks to river dolphins are more severe during the low water season when their habitats are fragmented and when dispersal ability is reduced [15]. Extinction of Yangtze River dolphins and a sharp decline in the populations of several river dolphins [Ganges River dolphin (*Platanista gangetica gangetica*), Indus River dolphin (*Platanista gangetica minor*), Irrawaddy dolphin (*Orcaella brevirostris*), Amazonian river dolphin (*Inia geoffrensis*), Bolivian river dolphin (*Inia boliviensis*), Tucuxi (*Sotalia fluviatilis*), Araguaian boto (*Inia araguaiaensis*)] are the example of consequences of inadequate flow in the Anthropocene [10, 11, 13, 16–18].

As habitat loss and fragmentation had significant adverse biological impacts on river cetaceans, previous studies recognized the sensitivity (presence/absence) of river dolphins to flow and their hydraulic properties (i.e., velocity, depth, and cross-sectional area) [19, 20]. Even though river flow directly regulates in-stream habitat structure that has a substantial impact on the organization and structure of biological communities [21], relationships between flow and ecological requirements of river cetaceans (particularly flows that determine the suitably usable area to river dolphins) have never been discussed and quantified. At the same time, appropriation of freshwater flows urged globally to sustain societal benefits and freshwater biodiversity by balancing the needs of humans, freshwater species, and ecosystems by setting limits for freshwater withdrawals and diversions [22]. Thus, there is an immediate need to understand the relationships between flow and river dolphins' hydro-physical requirements at a finer scale to understand the effects of flow alternation on in-stream habitat availability while balancing river dolphins' conservation and societal demands in the Anthropocene. Previous studies on aquatic ecosystem species conservation heavily consider ecological requirements of fish communities, thereby generalizing the flows over the entire ecosystem [23, 24]. Such an estimated

flow for fish communities may not be ecologically adequate and relevant to the species that are at the top trophic level with large body size, i.e., freshwater cetaceans. Furthermore, studies on habitat selection by freshwater cetaceans conservatively link the presence and abundance data to coarse-scale (~1–3 km) satellite image drove hydro-physical variables and visually characterize habitat structure [South Asian river dolphins [19, 20, 25], Irrawaddy water dolphin [19, 26], Amazonian river dolphin [15, 27], Bolivian river dolphin [28], Yangtze finless porpoise [29], Tucuxi [15] with hydro-physical habitat preference, however, do not assess the quality and quantity of the hydro-physical parameters at discrete points occupied by river cetaceans. Even though species are sensitive to certain flow velocities and depths [30], previous studies on freshwater cetaceans did not account for these parameters when examining the cetaceans' habitat preferences as these parameters play an important role in determining the availability of suitable in-stream habitat [20]. Lack of empirical relationships between river flow and river dolphins' ecological responses may further elevate extinction risks by limiting our ability to establish thresholds on environmental flows that sustain remaining riverine endangered cetaceans.

We quantify the relationships between flow and river cetacean ecology to determine the availability of suitable in-stream physical habitat as a response to flow. We tested the effect of the individual hydro-physical parameters (flow, depth, velocity, cross-sectional area, width, and wetted perimeter) on the availability of the in-stream usable habitat of the Ganges River dolphins (GRD) at fine spatial and temporal scale. We assumed that GRD prefers a particular combination (velocity and depth at a specific flow) of the abiotic environment that determines the quality and availability of in-stream physical habitat. We, thus, developed the Habitat Suitability Curve (HSC) using a combination of the depth and velocity preferred by GRD, which allows us to quantify the availability of suitable in-stream habitat in the form of area-weighted suitability (AWS, $m^2$/m). Our study quantifies the hydro-physical habitat selection of GRD at a wider level river flow, which could be essential to determine adequate river flows that are particularly significant for the conservation of in-stream habitats and critical to endangered freshwater cetaceans' life activities during the low water season. Thus, the results of this study are essential to balance aquatic species' conservation while minimizing the ecological risks induced through water extraction or diversion by development projects.

## Materials and methods

### Study area

This study was conducted in the Karnali River system of Nepal, where mega hydropower projects are under construction, and several are proposed and completed (Fig 1). Details about the current water-based projects and ecological significance are described in [10]. This river segment, particularly downstream, is highly significant to endanger aquatic species like GRD (*Platanista gangetica gangetica*), Gharial (*Gavialis gangeticus*), Smooth Indian otter (*Lutra perspicillata*), and diverse fish species. Current water extraction or diversion projects typically adopt a traditional biodiversity impact assessment (i.e., discussion and review based) but do not assess more detailed habitat relationships that include flow [31]. Additionally, water management plans adopted by such mega-projects are based on the proportion (or percentage) of total discharge at a particular season or time, undermining the aquatic species' ecological requirements across space and time. Thus, balancing aquatic species conservation and development is the foremost priority of the concerned stakeholders.

### Available habitat measurements

We conducted hydro-physical measurements across a 55 km continuous segment of the Karnali River of Nepal to the border with India during low water season when available habitat is

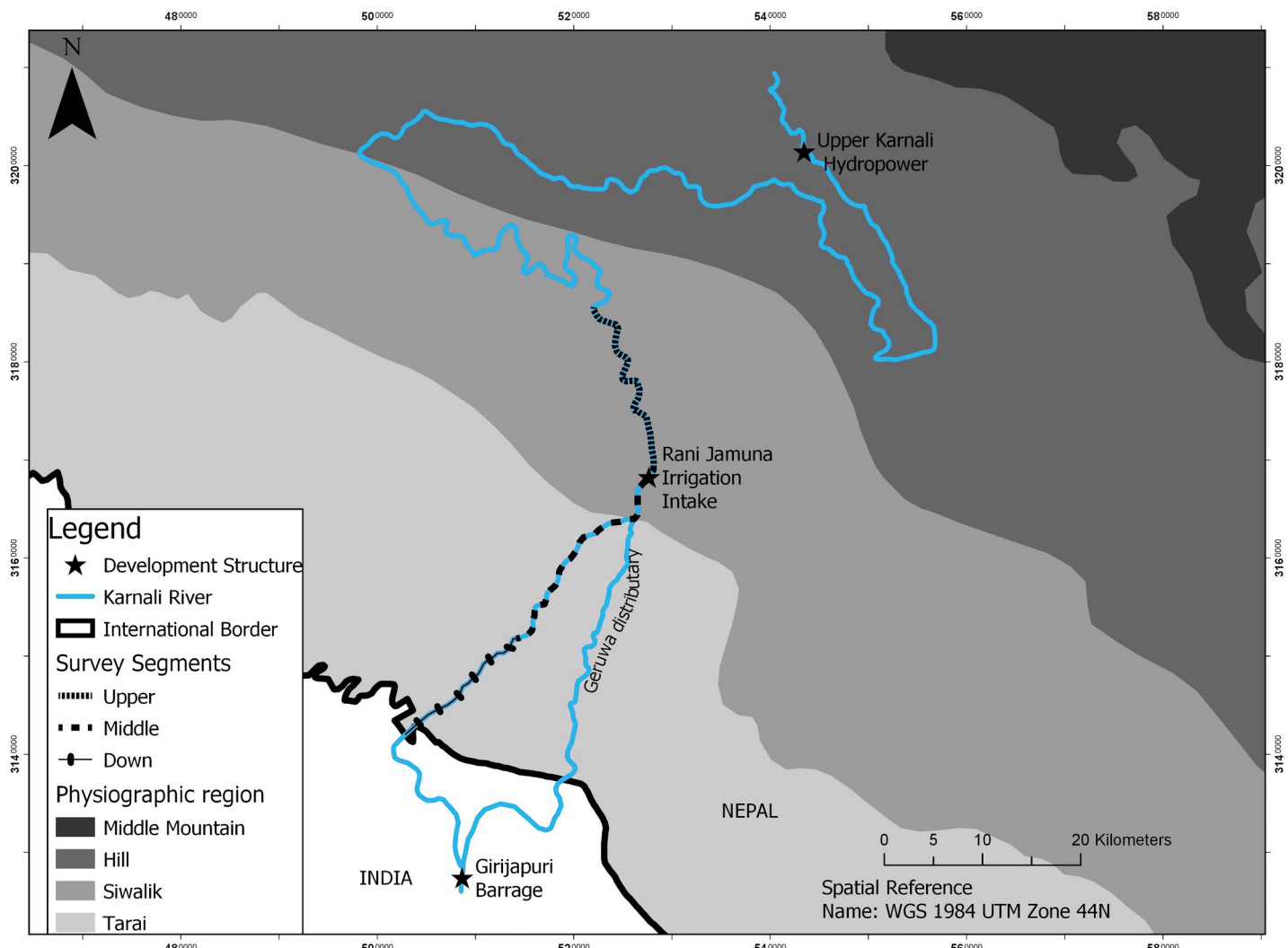

**Fig 1. A study map showing the location of development structures and segments took for hydro-physical measurements.**

reduced and river dolphin-anthropogenic activity interactions are intensified [32]. As the risk of changes in physical habitat may not truly represent the change in flow during the high flood season (or during daily fluctuation), we measured hydro-physical parameters two months from the termination of the monsoon (September). Field measurements of hydro-physical variables were conducted in December of 2017 and March and May of 2018 representing different flows (medium, low and high respectively) within the low water season. Therefore, our hydro-physical measurements represent a stable channel environment in response to flow. Flow was classified into different categories based on 39 years (1977–2015) of discharge data available for the study site from the Department of Hydrology, Government of Nepal. We stratified the study site into three segments (upper, middle, and lower, Fig 1) based on anthropogenic activities and hydro-physical characteristics of the river segment. The Mega hydropower project (Upper Karnali) is located upstream of the study site, whereas the Rani Jamara Kulariya Irrigation project water extraction point is in the upper segment of the study site. Within each segment, the study reach (the linear segment where cross-sections are established) was established in such a way that the length of each reach was at least higher than the mean width (so the

number varies among segments) of the respective segment. Further, we also tried to maintain a relatively similar flow at the top and bottom of the reach to create hydro-physically homogenous reaches. Within each reach, random cross-sections were established to capture the hydraulic properties based on flow variation and length of the reach.

We used SonTek RiverSurveyor S5 (Acoustic Doppler Current Profiler-ADP) systems to measure open channel hydro-physical parameters (total discharge-$m^3$/s; velocity-m/s; depth-m, wetted perimeter-m; total width-m, cross-sectional area (CSA)-$m^2$) at each cross-section, which used multiple acoustic frequencies along with precise bandwidth control for the most robust and continuous shallow-to-deep discharge measurements. An intelligent algorithm with high ping rates of S5 looks at water depth, velocity, and turbulence. It then acoustically adapts to those conditions to ensure robust data collection by compensating for vessel motion due to surface conditions. All discharge computations are collected within the S5. Total discharge at each cross-section, along with its attributes (velocity and depth) were measured at a cell size between 0.02 to 0.5 m, with potential measurement errors of 0.25% and 1% in velocity and depth, respectively. The ADP S5 hydraulic data were imported into Excel databases (Microsoft v. 2010) to format for System for Environmental Flow Analysis (SEFA, version 1.5; Aquatic Habitat Analysts Inc.) software, where we estimated AWS for each cross-section and reach using HSC (described in a later section). The habitat type (HT), i.e., pool, run, and riffle, was classified based on the Froude number (Fr), where Froude is an index of hydraulic turbulence (the ratio of velocity by the acceleration of gravity) [33]. Because this index is a simple habitat classification criterion based on hydraulic characteristics of these three habitats, and correctly classified 66% of the observed habitats in a similar riverine ecosystem, we applied this index as a basis for habitat classification. Points with Froude numbers exceeding 0.41 were considered riffles, points with Froude numbers less than 0.18 were considered pools and intermediate values were classified as run habitats [33]. Although this approach to habitat classification is tested in a different environment, we adopted the generally acceptable and simple approach to standardize the fieldwork using state of the art technology (ADP) as previous research relied on visual based classification, which undermines the hydro-morphological characteristics. Thus, we recommend further testing and verification of this habitat classification approach to fit to the local environment context before future applications.

We used basic descriptive statistics (mean, range, standard deviation (SD)) to characterize the hydro-physical parameters. Variation of these parameters across the season and habitat type was examined using F-test at a 5% level of significance. All these analyses were done in R-Studio version 1.3.95.

## GRD occupied habitat measurements

We measured the hydro-physical characteristics of the space used by GRD only in the main Karnali River, which is occupied by GRD most of the year. The tributary, Mohana segment, was excluded from the study as this tributary is only used by GRD during the monsoon season to avoid excessively high flow rates in the mainstream [32]. Also, across most of the year, this tributary exhibited seasonal characteristics (almost dry in the dry season). During each survey period, we used SonTek RiverSurveyor S5 for the measurement of hydro-physical parameters of GRD occupied. We used the widely applied Habitat Suitability Curve (HSC) approach to sustainable flow management while balancing aquatic conservation and societal demands to quantify area suitable to GRD [34]. We used the occupied habitat's hydro-physical parameters of depth and velocity to develop the Habitat Suitability Curve (HSC) using the procedure described in [10]. We derived a suitability curve as a function of depth and velocity, which allows quantification of the in-stream river dolphin's habitat availability in the form of area-

weighted suitability (AWS, m$^2$/m) at each cross section. Further, AWS of each cross-section is interpolated to get the value of AWS over the reach of the river [10]. This research was conducted under a research permit issued to the principal author of this paper by the Department of National Parks and Wildlife Conservation (DNPWC), Government of Nepal. All the observation procedures comply with regulations developed by the DNPWC.

## Area weighted suitability model

Because of complex relationships between response (AWS) and explanatory variables, the risk of violation of statistical assumptions (risk of non-homogeneity of the variance of a response variable, skewness in the distribution of response) is very high. To address this, we used generalized additive models for location, scale, and shape (GAMLSS) to build relationships between response (AWS) and explanatory variables (flow, velocity, depth, season (S), habitat type (HT), width, wetted perimeter, cross-sectional area (CSA)) which permits us to address those statistical issues within the model [35]. First, we selected the distribution of the response variable that can adequately describe the nature of the response variable by building linear models of explanatory variables using available potential distribution type (gamma, Box-Cox Cole-Green orig., normal, inverse gamma, lognormal) within the GAMLSS package. The model with proper distribution type was selected using generalized Akaike information criterion (GAIC). We found gamma distribution to be the best distribution family that can describe our response variable adequately. We used the gamma distribution for all our subsequent analysis.

We select significant model terms–linear, smoothing, and interactive–separately to develop the initial model that consists of all potential linear, smoothing, and interactive terms. A significant linear term was selected by developing a full linear model of seven explanatory variables. Then we used the *dropterm* function to get the final significant linear terms, which adopts stepwise model selection steps based on GAIC. To obtain the significant smoothing term, the smoothing parameters (except categorical variables) were selected by developing the initial null model, and then each smoother was added, one at a time, to the null model to get significant smoothing terms. Further, we determine whether two-way interactions are needed in the model using *stepGAIC* function in between intercept (lower model) and the most complicated model with all two-way interactions (upper). After having a full initial model with significant terms (liner, interactive, and smoothers), we again used the *dropterm* function to get the final model with the lowest GAIC.

We use *drop1* function available in the GAMLSS package to check for the approximate significance of the contribution of the smoothers (including the linear and interactive terms) at a 5% level of significance. To check the adequacy of the fitted GAMLSS model, we used a worm plot (*wp* function), which is a de-trended QQ-plot of the residuals [36]. For an adequate fitted model, we would expect the dots to be close to the middle horizontal line and 95% of them to lie between the upper and lower dotted curves, which act as 95% pointwise confidence intervals, with no systematic departure. All modeling procedures were completed using the *gamlss* package in R-Studio version 1.3.95.

## Results

### Available hydro-physical parameters

Available hydro-physical parameters differed over the season, except for depth (Table 1). All parameters were high in May and lowest in March. Across space or habitat type, hydro-physical parameters differ, except in flow (Table 2). The higher depth and cross-sectional area, along with the lowest velocity were recorded in deep pool habitat, followed by run and riffle habitats.

**Table 1. Hydro-physical parameters across seasons.**

| | December (N = 70) | March (N = 60) | May(N = 47) | Total (N = 177) | P.value |
|---|---|---|---|---|---|
| **Flow(m³/s)** | | | | | $F_{2,174} = 22.05, P < 0.001$ |
| Mean (SD) | 245.94 (81.52) | 203.14(64.56) | 326.33 (139.48) | 252.78 (106.73) | |
| Range | 82.96–420.34 | 81.52–475.61 | 50.60–665.49 | 50.60–665.49 | |
| **Depth(m)** | | | | | $F_{2,174} = 1.91, P = 0.15$ |
| Mean (SD) | 1.95 (0.79) | 1.92 (0.74) | 1.70 (0.47) | 1.87 (0.71) | |
| Range | 0.93–4.71 | 0.88–5.02 | 1.05–3.12 | 0.88–5.02 | |
| **Velocity(m/s)** | | | | | $F_{2,174} = 4.89, P = 0.009$ |
| Mean (SD) | 0.96 (0.33) | 0.83 (0.29) | 1.03 (0.35) | 0.94 (0.33) | |
| Range | 0.454–2.120 | 0.335–2.111 | 0.374–1.822 | 0.335–2.120 | |
| **Width(m)** | | | | | $F_{2,174} = 10.23, P < 0.001$ |
| Mean (SD) | 128.17 (46.54) | 126.18 (56.18) | 180 (102.37) | 141.26 (72.06) | |
| Range | 47.50–271.62 | 56.56–390.50 | 55.01–490.15 | 47.50–490.15 | |
| **CSA(m²)** | | | | | $F_{2,174} = 3.78, P = 0.025$ |
| Mean (SD) | 252.56 (136.35) | 233.31 (102.97) | 306.67 (181.26) | 260.40 (142.40) | |
| Range | 68.45–760.94 | 74.79–593.67 | 80.72–774.76 | 68.45–774.76 | |
| **WP(m)** | | | | | $F_{2,174} = 10.14, < 0.001$ |
| Mean (SD) | 129.29 (46.52) | 127.23 (55.96) | 180.87 (102.63) | 142.29 (72.06) | |
| Range | 48.25–272.68 | 57.61–391.19 | 57.13–491.93 | 48.25–491.93 | |

Variation on parameters across season tested using F-test and P value of significance reported.

**Table 2. Hydro-physical parameters across habitat type.**

| | Pool(N = 65) | Riffle(N = 20) | Run(N = 92) | Total(N = 177) | P.value (F-test) |
|---|---|---|---|---|---|
| **Flow(m³/s)** | | | | | $F_{2,174} = 0.36, 0.695$ |
| Mean (SD) | 261.78 (94.97) | 245.99 (97.43) | 247.90 (116.68) | 252.78 (106.73) | |
| Range | 50.60–481.55 | 121.34–475.61 | 95.68–665.49 | 50.60–665.49 | |
| **Depth(m)** | | | | | $F_{2,174} = 58.16, P < 0.001$ |
| Mean (SD) | 2.46 (0.80) | 1.39 (0.24) | 1.57 (0.35) | 1.87 (0.71) | |
| Range | 1.17–5.02 | 0.93–1.92 | 0.88–2.87 | 0.88–5.02 | |
| **Velocity(m/s)** | | | | | $F_{2,174} = 203, P < 0.001$ |
| Mean (SD) | 0.66 (0.14) | 1.59 (0.28) | 0.99 (0.18) | 0.94 (0.33) | |
| Range | 0.33–1 | 1.12–2.12 | 0.53–1.41 | 0.33–2.12 | |
| **Width(m)** | | | | | $F_{2,174} = 3.59, P = 0.03$ |
| Mean (SD) | 147.23 (64.87) | 101.22 (34.05) | 145.74 (80.27) | 141.26 (72.06) | |
| Range | 55.01–441.05 | 47.50–171.49 | 56.56–490.15 | 47.50–490.15 | |
| **CSA(m²)** | | | | | $F_{2,174} = 28.55, P < 0.001$ |
| Mean (SD) | 346.27 (139.31) | 139.07 (45.61) | 226.11(124.27) | 260.40 (142.40) | |
| Range | 80.72–760.94 | 68.45–246.03 | 74.79–774.76 | 68.45–774.76 | |
| **WP(m)** | | | | | $F_{2,174} = 3.61, P = 0.029$ |
| Mean (SD) | 148.73 (64.65) | 102.16(34.29) | 146.46 (80.35) | 142.29 (72.06) | |
| Range | 57.13–442.75 | 48.25–172.08 | 57.61–491.93 | 48.25–491.93 | |

Variation on parameters across season tested using F-test and P value of significance reported.

## GAMLSS model of area-weighted suitability

The model with the lowest GAIC value consists of the linear (flow, velocity, habitat type, season), smoother (depth, cross-sectional area) and interactive terms (depth and velocity, velocity and season, habitat type, and season) [Model 1, Table 3]. As indicated by the summary statistics of the quantile residuals of the best-fitted model (the mean of the model is nearly zero, variance almost one, the coefficient of skewness is almost zero, and coefficient of kurtosis is almost 3), the residuals are approximately normally distributed, as for an adequate model [34]. Furthermore, the Filliben correlation coefficient (or the normal probability plot correlation coefficient) is almost 1. 95% of the points lie between the two elliptic curves of the Worm plot (Fig 2). All these statistical and visual graphs suggest that the fitted distribution (or the fitted terms) of the model is adequate to explain the response variable.

Highly significant (the level of significance P < 0.001) linear predictors of the AWS were flow, velocity, season (March), depth, cross-sectional area, and of interactive terms were velocity and depth, and velocity and season (March). Flow and velocity exhibited linear negative relationships with AWS. In contrast, the depth and cross-sectional area revealed non-linear relationships, indicating both positive and negative contributions to AWS (Fig 3). Highest AWS was recorded in both December and March, and pools and riffle habitats exhibited

**Table 3. GAMLSS models with df (degree of freedom) and GAIC (generalized Akaike Information Criteria) for the response variable area-weighted suitability (AWS, $m^2$/m) and explanatory variables–flow ($m^3$/s), velocity (m/s), depth(m), habitat type (HT- deep pool, riffle, run), cross-sectional area ($m^2$) and season (December, March and May).**

| Model ID | Model | df | GAIC |
|---|---|---|---|
| 1 | **AWS ~ Flow + Velocity + HT + Season + pb(Depth)+pb(CSA) + Depth:Velocity + Velocity:Season + HT:Season** | 27.73 | 1258.52 |
| 2 | AWS ~ Flow + Depth + Velocity + CSA + HT + Season + pb(Depth) +pb(CSA) + Depth:Velocity + Flow:Season + Velocity:Season + HT:Season | 29.66 | 1262.49 |
| 3 | AWS ~ Flow + Depth + Velocity + CSA + HT + Season + pb(Depth) + Depth:Velocity + Flow:Season + Velocity:Season + HT:Season | 27.06 | 1276.33 |
| 4 | AWS ~ Flow + Depth + Velocity + CSA + HT + Season + Depth:Velocity +Velocity: Season + Depth:HT + Flow:CSA + HT:Season | 25 | 1286.28 |
| 5 | AWS ~ Flow + pb(Depth) + pb(Velocity) + CSA + HT + Season + Depth:Velocity + Flow:Season + Velocity:Season | 22.97 | 1290.27 |
| 6 | AWS ~ Flow + pb(Depth) + Velocity + CSA + HT + Season+Depth:Velocity + Flow: Season | 20.7 | 1292.98 |
| 7 | AWS ~ Flow + Depth + Velocity + CSA + HT + Season + Depth:Velocity + Velocity: Season + Depth:HT + Flow:CSA | 21 | 1300 |
| 8 | AWS ~ Flow + Depth + Velocity + CSA + HT + Season + Depth:Velocity +Flow:Season + Velocity:Season + Depth:HT + Flow:CSA | 23 | 1303.54 |
| 9 | AWS ~ Flow + Depth + Velocity + CSA + HT + Season + Depth:Velocity +Flow:Season + Velocity:Season + Depth:HT | 22 | 1309.94 |
| 10 | AWS ~ Flow + Depth + Velocity + CSA + HT + Season + Depth:Velocity + Flow: Season | 18 | 1313.73 |
| 11 | AWS~Flow + Depth + Velocity + CSA + HT + Season + Depth:Velocity +Flow:Season + Velocity:Season | 20 | 1315.13 |
| 12 | AWS ~ Flow + Depth + Velocity + CSA + HT + Season + Depth:Velocity | 16 | 1324.36 |
| 13 | AWS ~ Velocity + CSA + HT + Season | 13 | 1339.85 |
| 14 | AWS~Flow+Depth+Velocity+CSA+HT+Season | 15 | 1343.82 |

The best fitted model is highlighted in bold, which consist of linear, smoother (as indicated by pb), and interactive terms of the explanatory variables.

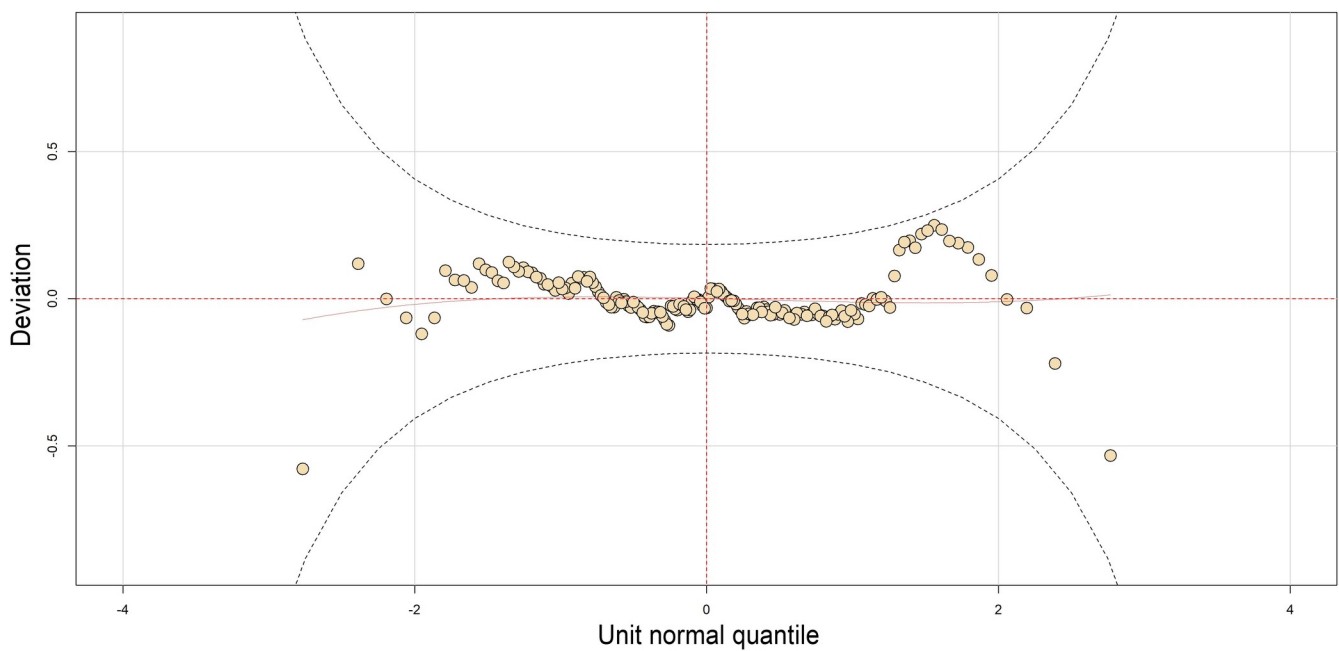

**Fig 2. A worm plot depicts that points of the worm plot are inside the 95% pointwise confidence intervals (curves) and close to the middle horizontal line, indicating adequacy of the fitted model.**

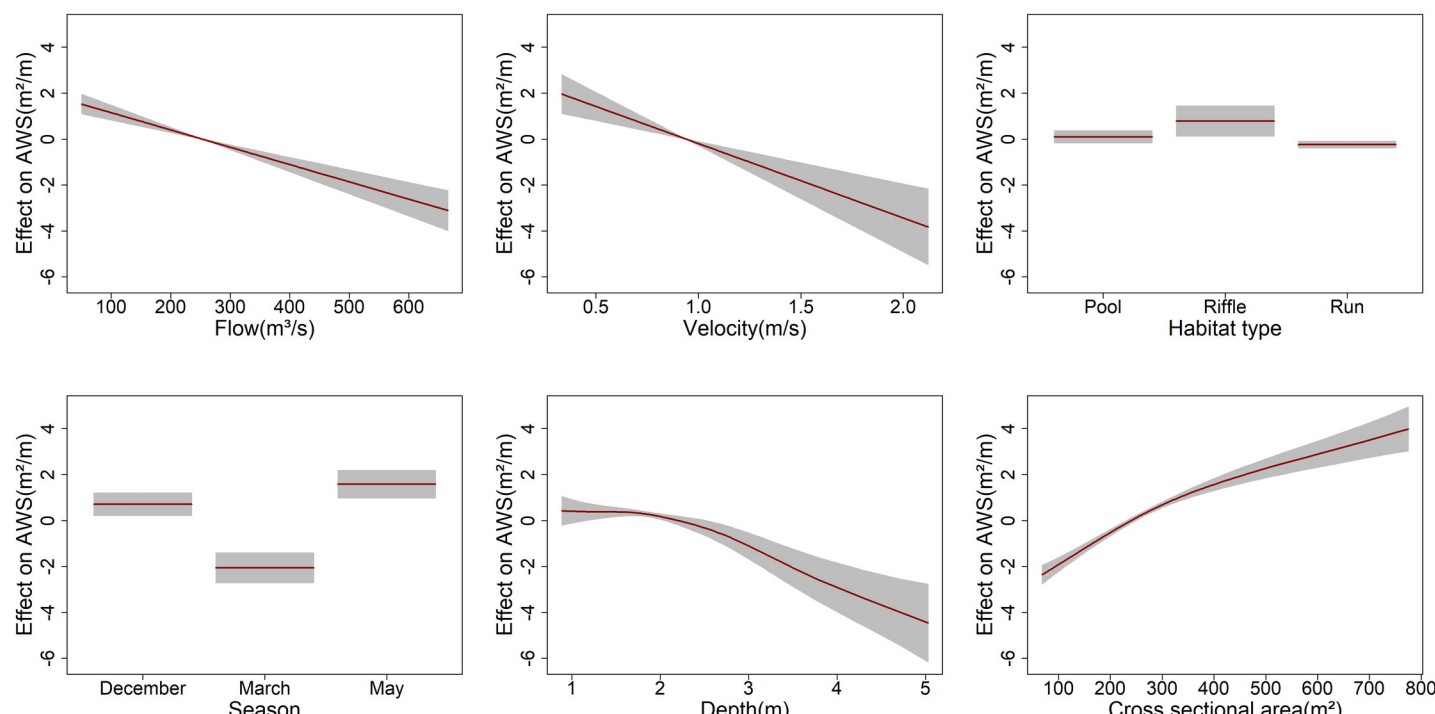

**Fig 3. A plot of the final fitted model shows that the predictor (AWS) declines linearly as flow and velocity increase, and AWS exhibited nonlinear relationships with depth and cross-sectional area.** The contribution of the season to AWS was higher for both December and May, whereas March contributed negatively to AWS. Deep pool and riffle contributed positively, but run habitat holds a negative contribution to AWS.

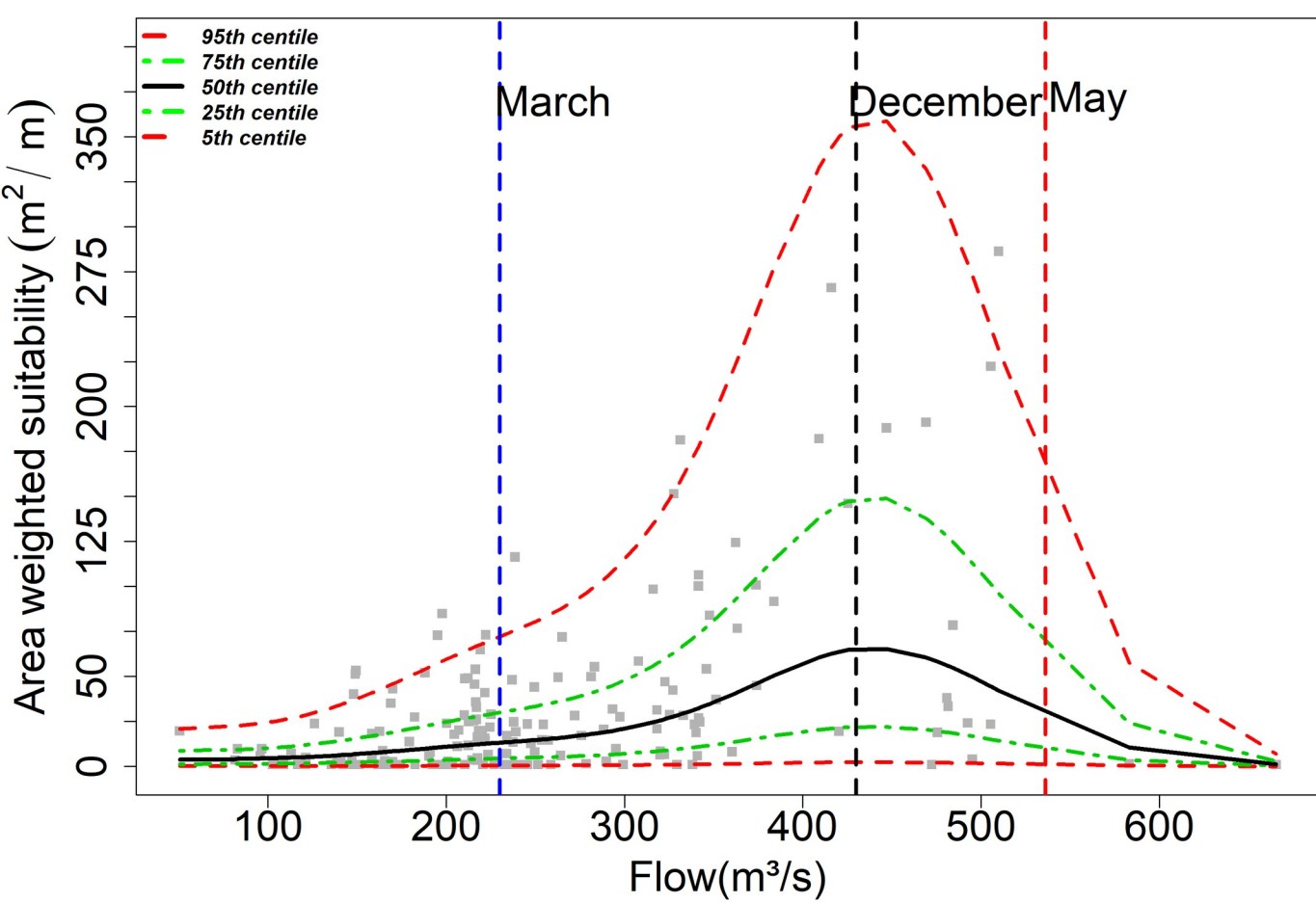

**Fig 4. Centile curves for AWS (m²/m) against the flow (m³/s) for different percentiles (5–95%) shows that maximum AWS is obtained with the flow around 450 m³/s. Vertical line represents an average of 90% exceedance flow of low (March), medium (December), and high (May) flow periods, suggesting highest AWS was available during May and AWS was substantially reduced during March and December due to reduced flow.** It indicates that the flow of March (peak dry season) needs to be protected as it is under critical flow to protect river dolphins, which contributes negatively to AWS.

higher AWS. The highest contribution to the variation of AWS was the interactions of the velocity and season (March) [β = 32.15, 95% CI range: 5.89–175.43], velocity and depth (β: 4.52, 95% CI: 2.26–9.03), followed by linear terms–cross-sectional area (β = 1.01, 95% CI: 1–1.01), flow (β = 0.992, 95% CI: 0.990–0.994), depth (β = 0.36, 95% CI: 0.21–0.60) and velocity (β = 0.03, 95% CI: 0.009–0.159) respectively. The cross-sectional area < 250 m² showed a negative contribution to the AWS, and above this point showed a positive linear contribution to the AWS (Fig 3). By season, peak dry season contributed negatively to the AWS. In contrast, both December and May had a positive contribution to the AWS in which the contribution of May was higher than December. Contribution of the deep pool and riffle habitats were higher to AWS than run (Fig 3). We model the contribution of wider flows (50–665 m³/s) to AWS, in which flows between 300 and 550 m³/s showed maximum positive contribution to AWS with the peak value of AWS at 450 m³/s (Fig 4). Relationships between flow and AWS reveal that a small and large value of flow has the same value of AWS. Velocity > 1m/s showed a negative contribution to the AWS in which 0.6 m/s had the highest positive contribution to AWS. Depth of 2–4 m maximized contribution to AWS, in which 2.5 m had the highest contribution to AWS (Fig 5).

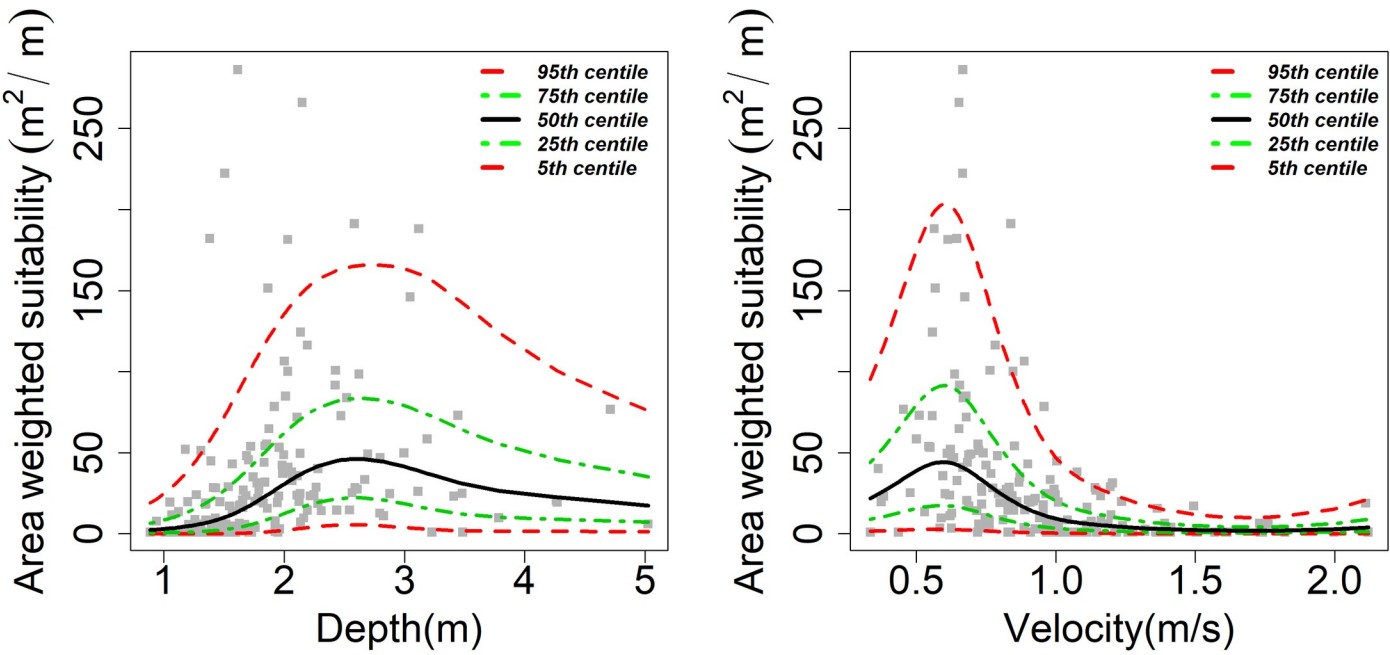

**Fig 5. A plot of centile curves for AWS (m²/m) fitted against depth and velocity separately indicates AWS is maximized when depth is between 2 and 3.5 m and velocity 0.3–0.8 m/s.** 95 centile value of AWS >150 m²/m when depth and velocity were 2.7 m and 0.6 m/s, respectively.

### Interactive effects of velocity and depth on area-weighted suitability

The velocity <0.25 m/s and depth with 2.5 m yielded the highest AWS (Fig 6). Depth between 2 and 3 m and higher cross-sectional area (>600 m²) predicted the highest AWS. Velocity between 0.5 and 1 m/s with a higher cross-sectional area (>600 m²) yielded the highest AWS.

## Discussion

Lack of information on the fine-scale habitat selection of large and mobile aquatic predators–river cetaceans–limits our ability to understand the broad-scale impacts of human-caused

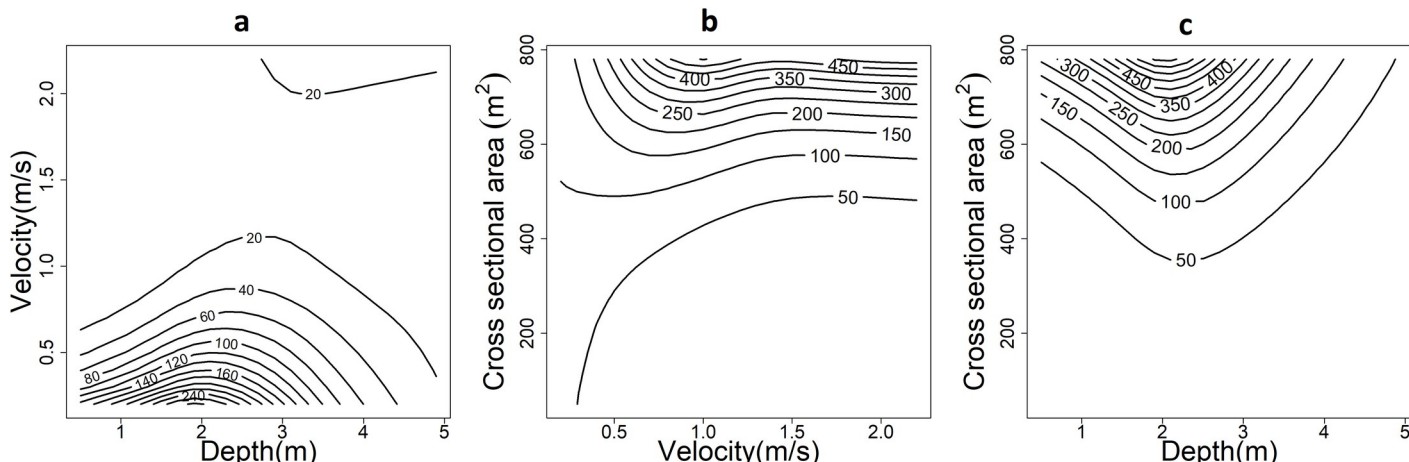

**Fig 6. A surface plot of the AWS (m²/m) showing the interactive effects of (a) velocity and depth, (b) cross-sectional area and velocity, (c) cross-sectional area, and depth.** The plot suggests that depth between 2 and 3 m combined with velocity <0.5 m/s maximized the AWS. Similarly, velocity and depth with a similar range along with a cross-sectional area >600 m² optimized the AWS.

changes to the dynamic riverine ecosystem. As river cetaceans exhibit a strong dependence on hydro-physical attributes, including geomorphic and seasonal flow variation effects, they are essential to informing the decision-making process of flow regulation management, including the identification and protection of river cetaceans' priority habitats. Our results can be used to incorporate knowledge about in-stream habitat selection and the potential effects of declining freshwater flow on the ecology of the large top predator of the riverine ecosystem into the design of habitat retention plan and water development projects that may alter the natural flow regime at the cost of aquatic species conservation. Incorporation of the ecology of a key mobile aquatic predator may help to conserve other lower trophic or similar aquatic species that share the river's physical habitat template and ecosystem [37].

Previous studies on river dolphins' habitat selection or preference commonly reported the aggregation of river dolphins in certain hydro-physical habitats [32, 38], especially deep pools. The majority of the studies attributed such habitat selection to the higher biological productivity and prey abundance [39, 40]. However, the ecological significance of such habitats to river dolphins has never been described explicitly. Here, we characterize the hydro-physical parameters of the deep pools that may be significant to the ecology of river dolphins. We demonstrate that deep pools offer higher cross-sectional areas combined with the deepest depth and considerably reduced water velocity, and in turn, yield the highest usable area to the river dolphins. The ecological significance of deep pools that offer the highest usable area is linked to the diving physiology of dolphins as the selection of deeper usable areas appears to be a mechanism that enables diving river cetaceans with limited oxygen stores to extend the duration of a dive [41]. Further, reduced velocity of water in deep pools in comparison to other habitats (run, riffle) may support river dolphins to reduce their swimming speed as the elevated swimming speeds during dive or swimming decreases the duration of a dive due to rapid depletion of limited oxygen reserves [42]. As a result, frequent diving in deep pools by river dolphins is ecologically and physiologically important by reducing the metabolic cost of dives. Such energetically efficient modes of locomotion in deep pools provide an advantage during periods of diving (submerge) and will presumably increase foraging efficiency as the animals perform progressively longer dives.

Global population decline and isolation risks of river dolphins is frequently attributed to declining flows due to dams or barrages [11, 16, 18, 32, 43, 44]. As a result, the need for adequate flow to conserve the remaining endangered river dolphins is heightened globally [10, 19, 20, 44, 45]. However, to our knowledge, information on quantitative relationships between flow and river cetaceans is scant, particularly for the low water season when habitat availability reduces, and longitudinal connectivity worsens. The linear negative relationships between flow and AWS combined with physical effects of flow variability in the cross-sectional area of the channel exemplified that immediate effects of flow fluctuation (either increase or decrease) are the loss of suitably usable area (i.e. habitat loss). All flows are not contributing in the same fashion to AWS, offering the opportunity to balance water extraction and conservation. Availability of AWS as a function of river flow shows that the flow of May (average 90% exceedance flow = 439.4 $m^3$/s) offers maximum AWS compared to the flows of December (average 90% exceedance flow = 335 $m^3$/s) and March (average 90% exceedance flow = 221 $m^3$/s) [Fig 4]. Flows in March represent critically low flow (negative contribution to AWS), which substantially reduces AWS. Flow from December to April seems more critical for river dolphin conservation. This flow variation across season suggests that flow is the major determinant of river cetaceans' hydro-physical habitat, which substantially alters the habitat at varying spatial and temporal scales. Further, the variation in flow is likely to control the migration strategies of aquatic species [46]. This is demonstrated by GRD in our study site. As the monsoon starts with the average 90% exceedance flow of 672 $m^3$/s in June, unusable habitats in the main

stream with flow >600 m$^3$/s (Fig 4) trigger GRD to migrate to a tributary (Mohana River; starts from the first week of June; personal observation), which offers seasonal refuge and suggests that a seasonal high flow has a major influence on shaping the life-history patterns of river dolphins. Similar high flow-regulated seasonal migration patterns are exhibited by river dolphins in an Amazonian floodplain [47]. As the river dolphins' birthing peak is during low water season [48], such high flow with the potential of physical disturbance immediately after the birth may pose a risk of recruitment failure for young calves [49].

The requirement of adequate flow is commonly cited to protect river dolphins and avoid the risk of population isolations that may cause recruitment failure and intensify local extinction [10, 11, 19, 20]. Our results may assist to determine the adequate flow across seasons that might have potential ecological significance to maintain the longitudinal connectivity essential to sustain the viability of river dolphins and other aquatic taxa that share the ecosystem. As water extraction and dam construction limits aquatic species dispersal ability to move freely through the stream network, extinction risk and endangerment of river cetaceans whose existence relies on flow will particularly be high during low water season. Our results could help river managers to predict how far in-stream usable areas can be altered from their natural state before water extraction or withdrawal occurs. As the species' abundance and distribution vary with stream depth and velocity [50], the microhabitat preferences of river cetaceans as a function of velocity and depth were poorly known. Our model shows that the distribution of suitable habitats (i.e. pools) of GRD is largely determined by the interactive effects of velocity and depth at a particular landform. The velocity (<0.5 m/s) combined with a depth between 2 and 3 m yielded a higher useable area to GRD in the study site (Fig 6). Further, a greater cross-sectional area (>600 m$^2$) with similar velocity and depth ranges offer maximum suitable areas to GRD. Such complex habitat selection behavior (depth, velocity, and cross-sectional area) exhibited by Ganges River dolphins in a dynamic aquatic environment indicates GRD are habitat specialists with narrow habitat breadth and environmental plasticity. Given the often-cited patchy distribution pattern of GRD [51], such interactive effects may play a significant role in the distribution and abundance of river cetaceans. Understanding river dolphins' specific in-stream habitat preference in relation to flow may offer an effective riverine conservation management plan for a heavily regulated river system [46].

Loss in habitat quality and the limited ability of species to migrate to new habitats is a major challenge for aquatic species conservation [52]. Our results indicate that the vulnerability of river cetaceans is increased by fragmentation and loss of longitudinal connectivity when the flow reduction is intensified either naturally or artificially. Higher risk of deep pool fragmentation and loss of longitudinal connectivity across suitable pools are commonly suggested conservation issues for river dolphins conservation, particularly in the dry season [10, 11, 17]. Effective flow regulation plans are critical to riverine cetacean conservation and management to avoid further acceleration of extinction risk [10, 11, 19, 20]. A 2D map of the segments (surface map of depth as a function of flow) for the low flow period (March) clearly indicates the substantial fragmentation in deep pools' (>2) availability along with potential dolphin passage barriers (depth <0.5m) as a response to reduce flow (Fig 7). Barriers to passage may limit GRD dispersal ability to suitable habitats, which may exacerbate the rate of endangerment and local extinction of the remaining populations of endangered GRD [53]. We suggest that conservation agencies must be adopted effective assessment procedures to determine the true impact of flow extraction or diversion projects to intensify conservation of aquatic biota in these vanishing riverine ecosystems.

The biological consequences of habitat loss and fragmentation in declining river cetaceans are evident. Births and mating activities of river dolphins generally peak during low water season when all animals are concentrated along the main stream [48], likely reducing pairing

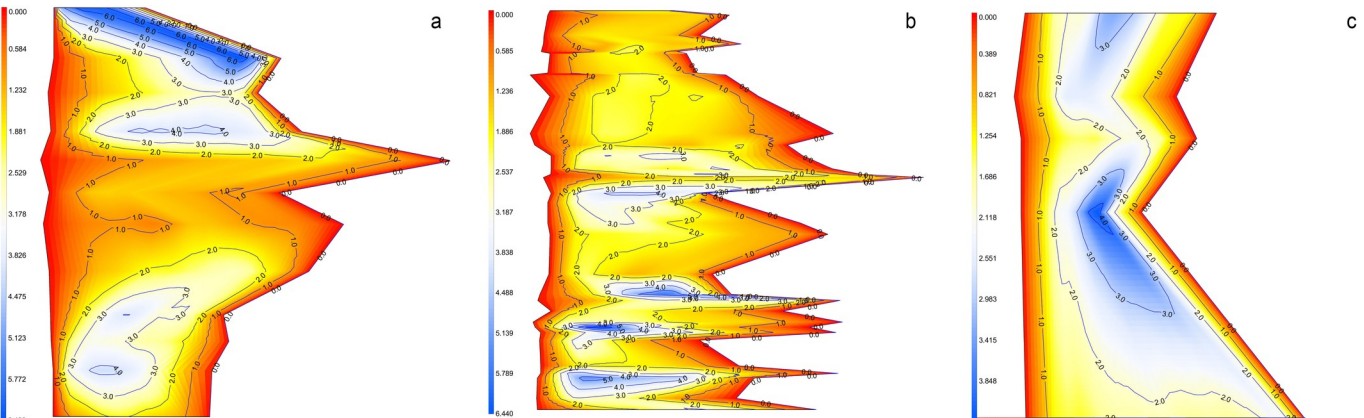

**Fig 7.** A pseudo-2D view of the depth (m) surface showing in contours and shading for segments (a) upper, (b) medium, (c) lower using interpolation technique. Light yellow indices depth <0.5 m and dark blue indicate deep pools >2 m, respectively. Sparsely located dark blue and substantially present light yellow over the continuum of the segments indicate acute habitat fragmentation and loss of connectivity across suitable pools during low water (peak dry season-March) when flow reduced.

success, which may hinder reproduction processes in isolated habitats and be a contributing factor to the low occupancy and decline of species, especially *Platanista gangetica* species. Further, spatial overlap during the low water season between river cetaceans and fisheries may intensify, leading to entanglement and reduction of survivorship [10, 14, 15]. Further diversion or withdrawal of flow from the mainstream contributes to habitat loss and undoubtedly leads to local extirpation. Combined effects of habitat loss with anthropogenic disturbances (fishing pressure and flow regulation by development projects) may lead to rapid population decline and local extinction of river cetaceans [53]. Rapid loss (~50%) of *Platanista gangetica* species is the clear evidence of such effects [10, 17]. Even though conserving the natural flow is unrealistic given societal demands [54], the seasonal effects and changes on the availability of usable areas to river cetaceans must be a central guiding element to conserve riverine biodiversity. Identifying priority riverscapes for river cetaceans, and prioritizing investment opportunities for development projects could be an essential first step towards effective conservation of riverine biodiversity.

Our research addresses a critical limitation in the ability to predict and quantify the river dolphins' response to flow regulation. We demonstrated quantitative relationships between flow and cetaceans ecology in the form of usable area availability in response to flow. These findings should permit the development of a proactive riverine cetacean management plan that incorporates habitat manipulation and protection, including the creation of migration/ movement corridors. Owing to the strong dependence on the attribute of freshwater flow, river dolphins are expected to be more affected by flow regulation as the interactive effects of flow and depth at particular landforms play a significant role to determine the usable area. Freshwater withdrawal and extraction by large-scale hydropower or irrigation projects will undoubtedly cause major ecological changes in the dynamic riverine ecosystem, which decreases the viability of river dolphins populations. River cetaceans are more sensitive to the attributes of freshwater flow, and the decline of freshwater flow may further put endangered cetaceans at risk of local extirpation, as exemplified by a sharp decline in abundance and range of GRD in a study site as flow regulation progress [32]. Knowledge of key hydro-physical habitat components may support development of a riverine habitat retention management plan in response to top predator or mega-species life history requirements [10].

## Acknowledgments

We thank Rajesh Sigdel, RC Gautam and various local stakeholders for assisting data collection process. Two reviewers provided helpful comments to increase the accessibility of the manuscript and we appreciate their efforts. Thanks to Department of National Parks and Wildlife Conservation, Government of Nepal for granting research permit.

## Author Contributions

**Conceptualization:** Shambhu Paudel, John L. Koprowski.

**Data curation:** Shambhu Paudel.

**Formal analysis:** Shambhu Paudel.

**Funding acquisition:** Shambhu Paudel.

**Investigation:** Shambhu Paudel, Usha Thakuri.

**Methodology:** Shambhu Paudel.

**Project administration:** Shambhu Paudel, John L. Koprowski, Usha Thakuri.

**Resources:** Shambhu Paudel.

**Software:** Shambhu Paudel.

**Supervision:** Shambhu Paudel, John L. Koprowski.

**Validation:** Shambhu Paudel.

**Visualization:** Shambhu Paudel.

**Writing – original draft:** Shambhu Paudel.

**Writing – review & editing:** John L. Koprowski, Usha Thakuri, Ajay Karki.

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
