## [Decision Letter · Decision Letter 0]

24 Nov 2020

PONE-D-20-31590

In-stream habitat availability for river dolphins in response to flow:  use of ecological integrity to manage river flows

PLOS ONE

Dear Dr. Paudel,

Thank you for submitting your manuscript to PLOS ONE. After careful consideration, we feel that it has merit but does not fully meet PLOS ONE’s publication criteria as it currently stands. Therefore, we invite you to submit a revised version of the manuscript that addresses the points raised during the review process.

I received two reviews for this manuscript.  Both reviewers commend the authors on the work and they think this type of research is needed urgently to further understand ways in which river dolphins can be conserved and viable populations can be maintained.  However, one of the reviewers, an expert in hydraulics, was critic considering the way in which the authors explain the concepts related to hydraulics. I cite him: "noted a few occasions when the authors did not appear to clearly understand what they were describing, and overall I felt that the presentation quality of the hydraulic results was low.  If possible I would suggest collaborating with someone in that field as the data set looks promising and the problem clearly merits attention.  I have made many other comments in the manuscript".  

The second reviewer, and expert in river dolphin conservation was very positive about the manuscript and considered acceptance.

Considering these two reviews, I suggest the authors that they thoroughly revise their manuscript, especially the concepts and results related with hydraulics.  Maybe communicate with a local expert in this field that can provide their expertise?  I personally consider that we need all the good quality information that can be made available for river dolphin conservation, so I really hope the authors work on this recommendation to improve the quality of their paper. 

We look forward to receiving your revised manuscript.

Kind regards,

Susana Caballero, PhD

Academic Editor

PLOS ONE

Additional Editor Comments:

I received two reviews for this manuscript. Both reviewers commend the authors on the work and they think this type of research is needed urgently to further understand ways in which river dolphins can be conserved and viable populations can be maintained. However, one of the reviewers, an expert in hydraulics, was critic considering the way in which the authors explain the concepts related to hydraulics. I cite him: "noted a few occasions when the authors did not appear to clearly understand what they were describing, and overall I felt that the presentation quality of the hydraulic results was low. If possible I would suggest collaborating with someone in that field as the data set looks promising and the problem clearly merits attention. I have made many other comments in the manuscript".

The second reviewer, and expert in river dolphin conservation was very positive about the manuscript and considered acceptance.

Considering these two reviews, I suggest the authors that they thoroughly revise their manuscript, especially the concepts and results related with hydraulics. Maybe communicate with a local expert in this field that can provide their expertise? I personally consider that we need all the good quality information that can be made available for river dolphin conservation, so I really hope the authors work on this recommendation to improve the quality of their paper.

Journal Requirements:

"Thanks to the numerous agencies -- Rufford Foundation (UK), WWF-EFN Program (USA),

 Institute of Forestry (Tribhuvan University, Nepal), Commonwealth Scientific and Industrial

Research Organization (CSIRO-Australia), WWF-Nepal, University of Arizona – School of

Natural Resources and the Environment – for field funding support."

"The author (SP) received small field research grant from: Rufford Foundation (UK), Commonwealth Scientific and Industrial Research Organization (CSIRO-Australia), and WWF-Nepal. The funders had no role in study design, data collection and analysis, decision to publish, or preparation of the manuscript."

3. Please amend your authorship list in your manuscript file to include author Ajay Karki.

4. We note that Figure 1 in your submission contain map images which may be copyrighted. All PLOS content is published under the Creative Commons Attribution License (CC BY 4.0), which means that the manuscript, images, and Supporting Information files will be freely available online, and any third party is permitted to access, download, copy, distribute, and use these materials in any way, even commercially, with proper attribution. For these reasons, we cannot publish previously copyrighted maps or satellite images created using proprietary data, such as Google software (Google Maps, Street View, and Earth). For more information, see our copyright guidelines: http://journals.plos.org/plosone/s/licenses-and-copyright.

41.    You may seek permission from the original copyright holder of Figure 1 to publish the content specifically under the CC BY 4.0 license. 

4.2.    If you are unable to obtain permission from the original copyright holder to publish these figures under the CC BY 4.0 license or if the copyright holder’s requirements are incompatible with the CC BY 4.0 license, please either i) remove the figure or ii) supply a replacement figure that complies with the CC BY 4.0 license. Please check copyright information on all replacement figures and update the figure caption with source information. If applicable, please specify in the figure caption text when a figure is similar but not identical to the original image and is therefore for illustrative purposes only.

Reviewers' comments:

Reviewer's Responses to Questions

**Comments to the Author**

1. Is the manuscript technically sound, and do the data support the conclusions?

Reviewer #1: No

Reviewer #2: Yes

2. Has the statistical analysis been performed appropriately and rigorously? 

Reviewer #1: I Don't Know

Reviewer #2: Yes

3. Have the authors made all data underlying the findings in their manuscript fully available?

Reviewer #1: No

Reviewer #2: Yes

4. Is the manuscript presented in an intelligible fashion and written in standard English?

Reviewer #1: Yes

Reviewer #2: Yes

5. Review Comments to the Author

Reviewer #1: I read this study with interest as it did describe a novel approach to understanding the habitat of river dolphins, which is critical, as the authors say, for understanding what the impacts of flow regulation in the form of hydropower megaprojects will be on habitat loss and fragmentation. As a researcher on the hydraulic side, this seemed to me an important opportunity to delve into an important issue of our time and to link it with channel hydraulics. I am not an expert on the AWS modelling and the statistical tools used to asses the model and so will not comment extensively on them. I commend the researchers on their efforts to obtain a unique and important dataset in this environment. However, the study should not be considered adequate from a hydraulic perspective. I noted a few occasions when the authors did not appear to clearly understand what they were describing, and overall I felt that the presentation quality of the hydraulic results was low. If possible I would suggest collaborating with someone in that field as the data set looks promising and the problem clearly merits attention. I have made many other comments in the manuscript, but overall my recommendation is to reject.

Reviewer #2: The manuscript is well prepared and develops the objectives proposed in the investigation in a clear and robust way.

Please change the term "riverine cetaceans" in the document to "river dolphins", and in the abstract section in the lines 30 -31 change the words "with reference to" for concerning or regarding. Introduction in the line 59 delete the words "the", in the line 67 delete ",", line 94 include "," in the phrase "species, and", line 103 "the" in the phrase "the presence", and change the word "driven" for drove or has driven, line 107 include "," the word "however", line 108 include "the" in the phrase "the quality", line 129 change the phrase " in relation to" for "about, to, with or concerning", and change the phrase "properties of wider level" for "properties of a wider level". Materials and Methods in the line 141 change the words "endangered" for "endanger", line 177 change the words "cross section" for "cross-section", line 192 remove "," of the phrase "considered pools, and intermediate", line 201 change the word "main stream" or "mainstream", line 202 change the word "characteristic" for "characteristics", lines 231 and 238 change the word "dropterm" for "drop term", lines 241 and 247 include "the" in the phrase " in the GAMLSS package to", and "were completed using the gamlss package". Results in the line 252 include the word "the season" or "a season", line 254 include the word "The higher", and "the lowest", line 275 include "," in the phrase " , and", line 278 include "the coefficient", and line 295 change the word "were flowed or were flowing". Discussion in the line 363 include "-" in the phrase "broad-scale", line 364 include "a" in the phrase "a strong", line 382 include " the river", line 390 change the word "is" for "are", line 413 include " with an average", line 436 delete "a" in the phrase "with similar velocity", line 438 change the word "indicates" or "indicate", line 446 in include the word "that the vulnerability", line 447 change the word "is" for "are" and include the word "the flow", line 488 include the word "the development", line 449 include "the viability", and line 496 change the word "under" for "at".

6. PLOS authors have the option to publish the peer review history of their article (what does this mean?). If published, this will include your full peer review and any attached files.

Reviewer #1: No

Reviewer #2: **Yes: **Federico Mosquera-Guerra Phd.

---

## [Author Response · Author response to Decision Letter 0]

29 Dec 2020

Date: 12-29-2020

Susana Caballero, PhD

Academic editor

PLOS ONE

Dear Susana,

First of all, I would like to thank you for the opportunity to revise and resubmit our manuscript, entitled “In-stream habitat availability for river dolphins in response to flow: use of ecological integrity to manage river flows”. I found the reviewers’ comments to be helpful in revising the manuscript and have carefully considered and responded to each suggestion. We completely addressed issues raised by the reviewer two but in the case of reviewer one, the reviewer seems like deviated from the core aim and approach of the paper due to technical limitations that connect ecology-flow. Since this paper is extended findings of the research which was recently published in Nature Scientific Reports (Ecological responses to flow variation inform river dolphin conservation | Scientific Reports (nature.com)) that builds flow-ecology relationships, the approach mentioned in the submitted paper to PLOS ONE is scientifically and technically validated (the same approach we published in Nature Scientific Reports was used). The reviewer mentioned technically inadequate, and recommend to consult the hydrologist. However, we had a team of biologists, hydrologist, geologist and engineers, who made this approach collectively to develop the flow-ecology relationships and now scientifically accepted as indicated by the publication in Nature Scientific Reports. Even though we mentioned clear measurement approach (available and occupied hydro-physical habitats) and links between flow-ecology through AWS calculation, the reviewer one finds hard to connect the links between flow-ecology. We added lines (199-203) and references [10, 33; above published paper] to make the approach and goal clearer to the reader. This is clearly mentioned in lines 118- 126 and 169-206. Again, the approach mentioned in this paper is already published that indicates technically and scientifically validated.

I have included a response to reviewers (below) in which we address each comment the reviewers made. In our response to reviewers, our responses follow immediately below the comment in yellow highlight. Corresponding changes are highlighted in yellow in the manuscript text in the revised file. 

Important notes:

1. Please include this funding statement in the system on behave of author: "The author (SP) received field research grants from: Rufford Foundation (UK), Commonwealth Scientific and Industrial Research Organization (CSIRO-Australia), and WWF-Nepal. The funders had no role in study design, data collection and analysis, decision to publish, or preparation of the manuscript."

2. Figure 1 is already published [ reference 10] written by same principal author (Shambhu Paudel). This figure is re-use under the CC BY License agreement.

Thank you again for your consideration of our revised manuscript.

 Sincerely,

Shambhu Paudel, Ph.D. scholar

School of Natural Resource and the Environment

University of Arizona, Tucson, Arizona, USA

And

Institute of Forestry, Pokhara, Nepal

Email: spaudel@email.arizona.edu

Response to reviewers:

Editor Comments:

I received two reviews for this manuscript. Both reviewers commend the authors on the work and they think this type of research is needed urgently to further understand ways in which river dolphins can be conserved and viable populations can be maintained. However, one of the reviewers, an expert in hydraulics, was critic considering the way in which the authors explain the concepts related to hydraulics. I cite him: "noted a few occasions when the authors did not appear to clearly understand what they were describing, and overall I felt that the presentation quality of the hydraulic results was low. If possible I would suggest collaborating with someone in that field as the data set looks promising and the problem clearly merits attention. I have made many other comments in the manuscript".

As this study combines river flow property and the ecology of the river dolphins to predict the in-stream habitat availability of river dolphins, the first reviewer finds it hard to understand its core approach and idea. Principally, we are not entirely working on modeling river hydraulics as does a hydrologist, but we quantify in-stream area availability to river dolphins as a function of river depth and flow at a particular flow level. To quantify habitat availability, we use ADP, which measures the complete cross-sectional dynamics using acoustics and provide us data on depth and velocity at cell size between 0.02-0.5 m, including cross-sectional area, width, discharge. Using hydro-physical properties of habitats with documented river dolphin use, we develop a habitat suitability curve, and then it was forwarded to estimate area-weighted suitability (AWS) measured in m2/m. AWS is the available suitable habitats at each cross-section. This details of the entire approach are described in lines 118-126 (introduction) and 169-206 (methods)

We also added a simple overview to the approach by adding lines 199-203 and a new reference in line 186.

The second reviewer, an expert in river dolphin conservation was very positive about the manuscript and considered acceptance.

The second reviewer who is river cetacean scientist did completely understand the approach that combines ecology and flow. We believe our additional details and a short overview should satisfy the breadth of readership that we suspect will find the paper useful. We appreciate the comments of Rev 1 and you to create a more accessible manuscript. Thank you!

Considering these two reviews, I suggest the authors that they thoroughly revise their manuscript, especially the concepts and results related with hydraulics. Maybe communicate with a local expert in this field that can provide their expertise? 

Our paper that details some of the methods was just published and we now cite that reference here in this PLoS ONE paper and that should further help:

Ecological responses to flow variation inform river dolphin conservation | Scientific Reports (nature.com)

We had a team of geologists, hydrologists and biologist, who designed the research approach, estimated the AWS and were instrumental in the general approach detailed in the Nature Sci Reports paper and applied in our manuscript submitted manuscript to PLoS ONE. We believe that our added details and the reference to the recent publication will assist with the accessibility of this work and hope that you agree!

I personally consider that we need all the good quality information that can be made available for river dolphin conservation, so I really hope the authors work on this recommendation to improve the quality of their paper.

As we already published the first outcome of this project, this paper, with the same approach, we elaborated further details about the river dolphin’s habitat in relation to the flow. The approach adopted by this paper was the same that we used in the research paper published in Nature Scientific reports (Ecological responses to flow variation inform river dolphin conservation | Scientific Reports (nature.com)). I hope this helps with the decision and ensures that we used a technically sound approach to make this paper scientifically robust.

Reviewer #1: I read this study with interest as it did describe a novel approach to understanding the habitat of river dolphins, which is critical, as the authors say, for understanding what the impacts of flow regulation in the form of hydropower megaprojects will be on habitat loss and fragmentation. As a researcher on the hydraulic side, this seemed to me an important opportunity to delve into an important issue of our time and to link it with channel hydraulics. I am not an expert on the AWS modelling and the statistical tools used to asses the model and so will not comment extensively on them.

The aim of this paper is to estimate AWS as a function of velocity and depth at the finer spatial resolution at a particular flow level. We estimated the AWS as each cross-section in response to flow level. It is the primary or core goal of this paper to inform practice and policy. We have added more details and an overview of the technique to assist with this understanding. 

I commend the researchers on their efforts to obtain a unique and important dataset in this environment. However, the study should not be considered adequate from a hydraulic perspective. 

We have changed some of our wording as mentioned above and provided more details as we are not particularly dealing with hydraulic modeling. Here, we combine flow (velocity and depth at particular flow level) with reference to dolphin requirements to develop flow-ecology relationships in terms of in-stream habitat availability (AWS). This is the goal of this paper … an application of the hydrological characteristics to the ecology and conservation of river dolphins. To complete this task, we measured a continuum available hydro-physical habitat (line 147) and occupied habitats by river dolphins (line 191) to develop the suitable habitat curve. This curve is further used to estimate AWS at each cross-section. Here we define river dolphin habitat by the interaction of velocity and depth (lines 118-120), which were measured by ADP at 0.02 to 0.5m spatial resolution across the cross-section. We tweaked the details of this approach in the methods section. This approach is the standard way to estimate habitat suitability; we have published the methods in greater detail elsewhere and now reference them as noted above. 

I noted a few occasions when the authors did not appear to clearly understand what they were describing, and overall I felt that the presentation quality of the hydraulic results was low.

We added further lines (199-203) and two references [10, 33] to make approach clearer to the reader.

If possible I would suggest collaborating with someone in that field as the data set looks promising and the problem clearly merits attention.

This is a project with a team member representing diverse areas, from ecologist, biologist, geologist, hydrologist to engineers. We have published the details elsewhere as noted above but we add more detail and citation to assist with the reader with the approach.

I have made many other comments in the manuscript,

We accepted all the comments made in the manuscript, which are highlighted by yellow color.

 but overall my recommendation is to reject.

I think we have made the approach clearer, added information and references to improve the quality of the paper.

Reviewer #2: The manuscript is well prepared and develops the objectives proposed in the investigation in a clear and robust way.

Thank you! Yes, we appreciate that you found the details clear and the manuscript valuable.

Please change the term "riverine cetaceans" in the document to "river dolphins", and in the abstract section in the lines 30 -31 change the words "with reference to" for concerning or regarding. Introduction in the line 59 delete the words "the", in the line 67 delete ",", line 94 include "," in the phrase "species, and", line 103 "the" in the phrase "the presence", and change the word "driven" for drove or has driven, line 107 include "," the word "however", line 108 include "the" in the phrase "the quality", line 129 change the phrase " in relation to" for "about, to, with or concerning", and change the phrase "properties of wider level" for "properties of a wider level". Materials and Methods in the line 141 change the words "endangered" for "endanger", line 177 change the words "cross section" for "cross-section", line 192 remove "," of the phrase "considered pools, and intermediate", line 201 change the word "main stream" or "mainstream", line 202 change the word "characteristic" for "characteristics", lines 231 and 238 change the word "dropterm" for "drop term", lines 241 and 247 include "the" in the phrase " in the GAMLSS package to", and "were completed using the gamlss package". Results in the line 252 include the word "the season" or "a season", line 254 include the word "The higher", and "the lowest", line 275 include "," in the phrase " , and", line 278 include "the coefficient", and line 295 change the word "were flowed or were flowing". Discussion in the line 363 include "-" in the phrase "broad-scale", line 364 include "a" in the phrase "a strong", line 382 include " the river", line 390 change the word "is" for "are", line 413 include " with an average", line 436 delete "a" in the phrase "with similar velocity", line 438 change the word "indicates" or "indicate", line 446 in include the word "that the vulnerability", line 447 change the word "is" for "are" and include the word "the flow", line 488 include the word "the development", line 449 include "the viability", and line 496 change the word "under" for "at".

We appreciate the suggestions and make all changes highlighted in yellow color in the manuscript and agree that the readability is improved.

---

## [Decision Letter · Decision Letter 1]

26 Apr 2021

PONE-D-20-31590R1

In-stream habitat availability for river dolphins in response to flow:  use of ecological integrity to manage river flows

PLOS ONE

Dear Dr. Paudel,

Thank you for submitting your manuscript to PLOS ONE. After careful consideration, we feel that it has merit but does not fully meet PLOS ONE’s publication criteria as it currently stands. Therefore, we invite you to submit a revised version of the manuscript that addresses the points raised during the review process.

both reviewers were positive regarding the contents and interest of your manuscript.  However, one of the reviewers considers more work needs to be done to explain some of your methods and the statistical analyses you did.  One of the reviewers noted that the authors may have had a defensive position when answering to the reviewer queries or suggestions.  Remember that science is done by presenting our work to colleagues (pair review) and what this approach wants to achieve is to be able to look at our work under different eyes and being able to be accept criticism and suggestions to improve the clarity and quality of our work.  I am sure you will be able to improve these aspect of your manuscript in the next version of it!!!

Please submit your revised manuscript by  1st of June 2021. If you will need more time than this to complete your revisions, please reply to this message or contact the journal office at plosone@plos.org. Please include the following items when submitting your revised manuscript:

We look forward to receiving your revised manuscript.

Kind regards,

Susana Caballero, PhD

Academic Editor

PLOS ONE

Additional Editor Comments (if provided):

both reviewers were positive regarding the contents and interest of your manuscript. However, one of the reviewers considers more work needs to be done to explain some of your methods and the statistical analyses you did. One of the reviewers noted that the authors may have had a defensive position when answering to the reviewer queries or suggestions. Remember that science is done by presenting our work to colleagues (pair review) and what this approach wants to achieve is to be able to look at our work under different eyes and being able to be accept criticism and suggestions to improve the clarity and quality of our work. I am sure you will be able to improve these aspect of your manuscript in the next version of it!!!

Reviewers' comments:

Reviewer's Responses to Questions

**Comments to the Author**

1. If the authors have adequately addressed your comments raised in a previous round of review and you feel that this manuscript is now acceptable for publication, you may indicate that here to bypass the “Comments to the Author” section, enter your conflict of interest statement in the “Confidential to Editor” section, and submit your "Accept" recommendation.

Reviewer #1: (No Response)

Reviewer #2: All comments have been addressed

2. Is the manuscript technically sound, and do the data support the conclusions?

Reviewer #1: No

Reviewer #2: Yes

3. Has the statistical analysis been performed appropriately and rigorously? 

Reviewer #1: No

Reviewer #2: Yes

4. Have the authors made all data underlying the findings in their manuscript fully available?

Reviewer #1: Yes

Reviewer #2: Yes

5. Is the manuscript presented in an intelligible fashion and written in standard English?

Reviewer #1: Yes

Reviewer #2: Yes

6. Review Comments to the Author

Reviewer #1: A few comments on the revision:

1. At line 183, I’m still not sure where this definition of the Froude number comes from. You may be thinking of the Reynolds number, which is a better index of turbulence because it is the ratio of the inertial to the viscous force. Froude number is a ratio of the inertial to the gravitational force and is an index of wave behaviour. In rivers it is used to understand transitions between sub and supercritical flow. This poor definition of the Froude number was one of the main reasons I suggested working with someone with a better knowledge of open channel flow hydraulics.

2. At line 186 it is good to see that a reference is now provided to indicate the source for these Froude number thresholds. I note, however, that these numbers are from a single braided river in the Alps (Jowett, 1993). They are applied uncritically in this case to a much different river in terms of depth and velocity. I very much doubt that the morphologic description of riffles and pools from shallow braided river would apply to this much larger river where dolphins reside. The values used for the Froude number based classification system is thus still not well justified.

3. For the GAMLSS models and results presented in Table 3, the Habitat Type (HT) is based on Froude number, which is based on Velocity and Depth. What is the benefit or logic of including Velocity Depth and Habitat Type on the model? Can't you do it with Velocity and Depth alone? Based on Figure 3, is habitat type really useful? It still seems to me that you are double counting or mixing hydraulic variables. Riffles have high velocity, but low depth, which according to the other plots have opposite effects on AWS. Did you test any models without HT? In Table 3 they all have it.

4. Figure 7 is still poor quality. There no axes labels, labels on the color bars, or indications of what the contours mean and the contour labels are too small to read.

5. In the response to my comments the authors state that they are ‘not particularly dealing with hydraulic modeling’, but they are using hydraulic measurements and concepts and using those to describe the habitat of these river dolphins. In offering my opinion I am not deviating from the core aim of your paper as you state in the letter, but simply trying to ensure that sound science is being published. I am judging the work solely based on what is in the current paper and have not consulted the Nature paper, which, while a significant accomplishment, is not the subject of the current review. The manuscript in front of me has some serious flaws related to the core idea that you are describing the hydraulic habitat of these creatures and these have not been addressed by your revision.

Reviewer #2: he recommendations suggested in the first stage of revision have been incorporated into the manuscript. Besides, it is an important contribution to the knowledge of threats to river dolphins in the Asian continent.

7. PLOS authors have the option to publish the peer review history of their article (what does this mean?). If published, this will include your full peer review and any attached files.

Reviewer #1: No

Reviewer #2: **Yes: **Federico Mosquera-Guerra PhD.

---

## [Author Response · Author response to Decision Letter 1]

22 May 2021

Reviwer#1:

1. At line 183, I’m still not sure where this definition of the Froude number comes from. You may be thinking of the Reynolds number, which is a better index of turbulence because it is the ratio of the inertial to the viscous force. Froude number is a ratio of the inertial to the gravitational force and is an index of wave behaviour. In rivers it is used to understand transitions between sub and supercritical flow. This poor definition of the Froude number was one of the main reasons I suggested working with someone with a better knowledge of open channel flow hydraulics.

Thank you for your suggestion. We follow Jowett (1993-A method for objectively identifying pool, run, and riffle habitats from physical measurements) and System for Environmental Flow Analysis (SEFA; Jowett et al. 2019) software manual to classify the habitats into the pool, run and riffle habitats from physical measurements taking the Froude number as an index of hydraulic turbulence. The reference papers compare the hydraulic characteristics of these three habitats and derive a simple habitat classification criterion which we applied in our study. This is a simple classification rule which correctly classified 66% of the habitats (Jowett, 1993). We have added some of these statements (including reference) to the methods section (lines 184-187) and then the reviewer for requesting more clarity.

2. At line 186 it is good to see that a reference is now provided to indicate the source for these Froude number thresholds. I note, however, that these numbers are from a single braided river in the Alps (Jowett, 1993). They are applied uncritically in this case to a much different river in terms of depth and velocity. I very much doubt that the morphologic description of riffles and pools from shallow braided river would apply to this much larger river where dolphins reside. The values used for the Froude number-based classification system is thus still not well justified.

As this is the first study in the region that develops flow-ecology relationships taking behavioral ecology of a large predator, the adopted approaches can be revised or revisited in the future study as this approach correctly classified 66% of the habitats. Previous studies visually classified the habitat types; however, we consider hydraulic properties measured by ADP at a higher spatial scale to classify the habitat. At the initial phase of this project, we consulted our approach with a senior hydrologists and key stakeholders in Nepal who are responsible for aquatic species and flow management before conducting field sampling. Thus, we have adopted the generally acceptable and simple approach to standardize the fieldwork using state-of-the-art technology so that these can be applied by biologists in the field. We appreciate the reviewer’s concern and have added a statement about the need for further testing and refinement of habitat classification in the future (lines: 189-194).

3. For the GAMLSS models and results presented in Table 3, the Habitat Type (HT) is based on Froude number, which is based on Velocity and Depth. What is the benefit or logic of including Velocity Depth and Habitat Type on the model? Can't you do it with Velocity and Depth alone? Based on Figure 3, is habitat type really useful? It still seems to me that you are double counting or mixing hydraulic variables. Riffles have high velocity, but low depth, which according to the other plots have opposite effects on AWS. Did you test any models without HT? In Table 3 they all have it.

First, we develop the full linear model using all the available variables to avoid bias on parameter selection. Then to get the final model with only significant terms, stepwise model selection step applied which helps to select the top model based on GAIC. This indicates parameters are included or excluded systematically based on significant contribution to the response variable (AWS).

After having these linear, interactive, and smoother terms which develop the full initial model, we again use a stepwise selection process based on GAIC. All models listed in the table are developed following the modeling procedure that considers only significant contributor and removes non-relevant elements by the stepwise process systematically. 

We explicitly described all the statistical steps described above or that we followed in the lines 219-253.

4. Figure 7 is still poor quality. There no axes labels, labels on the color bars, or indications of what the contours mean and the contour labels are too small to read.

We included a high-quality image with label and color bar in the revised manuscript. We submitted journal standard quality, however, while creating a PDF for review, it loses it's quality. We uploaded high quality individual file of each maps separately in the system for the final publication.

5. In the response to my comments the authors state that they are ‘not particularly dealing with hydraulic modeling’, but they are using hydraulic measurements and concepts and using those to describe the habitat of these river dolphins. In offering my opinion I am not deviating from the core aim of your paper as you state in the letter, but simply trying to ensure that sound science is being published. I am judging the work solely based on what is in the current paper and have not consulted the Nature paper, which, while a significant accomplishment, is not the subject of the current review. The manuscript in front of me has some serious flaws related to the core idea that you are describing the hydraulic habitat of these creatures and these have not been addressed by your revision.

This paper aims to quantify in-stream habitat availability using the predominantly applied Habitat Suitability Curve (HSC) approach (see application and relevance of HSC in aquatic ecology- Nestler et al. 2019). This approach is one of several approaches used in sustainable flow management while balancing societal demands and conservation. We built the HSC (based on velocity and depth) to estimate AWS (area-weighted suitability), the steps to develop HSC are explicitly described in Paudel et al. 2020 and cited in this paper as well to facilitate the learning process for the audience. Only the interactive function of velocity and depth was used to estimate AWS. We applied a widely consider approach to quantify the in-stream habitat availability in response to flow, but we did not include habitat type while estimating the AWS. As far as your concern is on the application of Froude number for habitat classification, habitat type was only included in the modeling process while identifying significant linear and interactive terms that define the AWS. Thus, we adopted an accepted approach while estimating AWS that includes only velocity and depth parameters. We believe that our modeling and quantification are explicitly defined and systematically conducted. Thus, we believe that our approach adopts simple mathematical steps that are justifiable and acceptable in the flow management process. Methodology with this clarification is now updated in lines (206-212).

Reviewer #2: The recommendations suggested in the first stage of revision have been incorporated into the manuscript. Besides, it is an important contribution to the knowledge of threats to river dolphins in the Asian continent.

Thank you!

---

## [Decision Letter · Decision Letter 2]

5 Jul 2021

In-stream habitat availability for river dolphins in response to flow:  use of ecological integrity to manage river flows

PONE-D-20-31590R2

Dear Dr. Paudel,

We’re pleased to inform you that your manuscript has been judged scientifically suitable for publication and will be formally accepted for publication once it meets all outstanding technical requirements.

Kind regards,

Susana Caballero, PhD

Academic Editor

PLOS ONE

Additional Editor Comments (optional):

Both reviewers suggested your manuscript to be accepted for publication. The only suggestion they made is to try to improve the quality of figure 7 (resolution) as well as the designation of the axis.

Reviewers' comments:

Reviewer's Responses to Questions

**Comments to the Author**

1. If the authors have adequately addressed your comments raised in a previous round of review and you feel that this manuscript is now acceptable for publication, you may indicate that here to bypass the “Comments to the Author” section, enter your conflict of interest statement in the “Confidential to Editor” section, and submit your "Accept" recommendation.

Reviewer #1: (No Response)

Reviewer #2: All comments have been addressed

2. Is the manuscript technically sound, and do the data support the conclusions?

Reviewer #1: No

Reviewer #2: Yes

3. Has the statistical analysis been performed appropriately and rigorously? 

Reviewer #1: Yes

Reviewer #2: Yes

4. Have the authors made all data underlying the findings in their manuscript fully available?

Reviewer #1: Yes

Reviewer #2: Yes

5. Is the manuscript presented in an intelligible fashion and written in standard English?

Reviewer #1: Yes

Reviewer #2: Yes

6. Review Comments to the Author

Reviewer #1: The authors have made some attempt to address my comments, although I would still disagree with their description of the Froude number and have not shown me a good quality version of their Figure 7, which I found difficult to read. Their responses overall are acceptable and the paper will likely be of interest to a wide audience.

Reviewer #2: The research is solid and robust, and the manuscript will surely be considered as a reference in the study of Asian river dolphins.

7. PLOS authors have the option to publish the peer review history of their article (what does this mean?). If published, this will include your full peer review and any attached files.

Reviewer #1: No

Reviewer #2: **Yes: **Federico Mosquera Guerra

---

## [Editor Report · Acceptance letter]

8 Jul 2021

PONE-D-20-31590R2 

In-stream habitat availability for river dolphins in response to flow:  use of ecological integrity to manage river flows 

Dear Dr. Paudel:

I'm pleased to inform you that your manuscript has been deemed suitable for publication in PLOS ONE. Congratulations! Your manuscript is now with our production department. 

Kind regards, 

on behalf of

Dr. Susana Caballero 

Academic Editor

PLOS ONE